# Immune Mechanisms Underlying Susceptibility to Endotoxin Shock in Aged Hosts: Implication in Age-Augmented Generalized Shwartzman Reaction

**DOI:** 10.3390/ijms20133260

**Published:** 2019-07-02

**Authors:** Manabu Kinoshita, Masahiro Nakashima, Hiroyuki Nakashima, Shuhji Seki

**Affiliations:** Department of Immunology and Microbiology, National Defense Medical College, 3-2 Namiki, Tokorozawa, Saitama 359-8513, Japan

**Keywords:** LPS, septic shock, elderly, innate lymphocytes, endotoxin tolerance

## Abstract

In recent decades, the elderly population has been rapidly increasing in many countries. Such patients are susceptible to Gram-negative septic shock, namely endotoxin shock. Mortality due to endotoxin shock remains high despite recent advances in medical care. The generalized Shwartzman reaction is well recognized as an experimental endotoxin shock. Aged mice are similarly susceptible to the generalized Shwartzman reaction and show an increased mortality accompanied by the enhanced production of tumor necrosis factor (TNF). Consistent with the findings in the murine model, the in vitro Shwartzman reaction-like response is also age-dependently augmented in human peripheral blood mononuclear cells, as assessed by enhanced TNF production. Interestingly, age-dependently increased innate lymphocytes with T cell receptor-that intermediate expression, such as that of CD8^+^CD122^+^T cells in mice and CD57^+^T cells in humans, may collaborate with macrophages and induce the exacerbation of the Shwartzman reaction in elderly individuals. However, endotoxin tolerance in mice, which resembles a mirror phenomenon of the generalized Shwartzman reaction, drastically reduces the TNF production of macrophages while strongly activating their bactericidal activity in infection. Importantly, this effect can be induced in aged mice. The safe induction of endotoxin tolerance may be a potential therapeutic strategy for refractory septic shock in elderly patients.

## 1. Introduction

Severe sepsis, particularly Gram-negative sepsis, remains a grave concern for humans, being recognized as a major leading cause of death in the United States as well as other countries [1]. Gram-negative septic shock, namely endotoxin shock, shows a particularly high mortality rate despite recent advances in intensive care. For several decades, the elderly populations has been rapidly increasing in many countries, and such patients are susceptible to endotoxin shock [2]. Therefore, medical countermeasures for this population are an urgent issue to be addressed. The generalized Shwartzman reaction is well recognized as an experimental endotoxin shock. We investigated the effect of aging on host defenses against endotoxin shock using a mouse model and human peripheral blood mononuclear cells (PBMCs) as an in vitro model of a Shwartzman reaction-like response, focusing on age-related immunological changes. We found that a certain T lymphocyte population that is age-dependently increased in mice as well as in humans collaborates with macrophages and causes the exacerbation of endotoxin shock in elderly subjects. In this review, we demonstrate why elderly hosts are susceptible to endotoxin shock, based on our immunological studies in murine and human PBMCs, and we also address potential therapeutic approaches against endotoxin shock in the elderly.

## 2. Definition of Septic Shock

Septic shock is the most severe form of sepsis and remains a grave concern because of its high mortality rate [3,4]. In 1991, the consensus conference of American College of Chest Physicians/Society of Critical Care Medicine defined sepsis as a systemic inflammatory response syndrome (SIRS) to infection [5,6]. Severe sepsis was also defined in that consensus conference as instances in which sepsis was complicated by acute organ dysfunctions, and septic shock was defined as sepsis complicated by either hypotension that was refractory to fluid resuscitation or by hyperlactatemia indicating tissue hypoxia [5,6]. At the time, the criteria for SIRS were thought to be clinically useful as a screening tool for the diagnosis of septic patients, as it can easily and simply define the severe infection and then effective therapy can be promptly started for SIRS conditions. In addition, these definitions were thought to be helpful for establishing patient entry criteria in clinical trials on sepsis.

One decade later, a second consensus conference was held in 2001 that endorsed most of these concepts, but some caveats were raised: Signs of SIRS (tachycardia, increased white blood cell count, fever elevation, tachypnea, etc.) may occur in not only infectious but also noninfectious conditions, so these criteria of SIRS may not always be useful for distinguishing sepsis from other critical conditions [7].

In 2016, a third consensus conference was held that defined ‘Sepsis-3′ [8]. In that conference, ‘sepsis’ was defined as life-threatening organ dysfunctions caused by a dysregulated host response to infection, similar to the definition of ‘severe sepsis’ proposed at the first conference in 1991. Septic shock was also defined as a subset of sepsis in which particularly profound circulatory, cellular, and metabolic abnormalities were associated with a greater risk of multi-organ dysfunctions and mortality than with sepsis alone [8].

## 3. High Mortality and Morbidity of Septic Shock in the Elderly

Aging is one of the most representative risk factors of septic shock [9]. Elderly people tend to have various kinds of chronic comorbid illness, such as ischemic heart disease, cerebrovascular disease, diabetes, and renal or pulmonary disease. These comorbid illnesses are presumably susceptible to sepsis/severe sepsis, leading to shock and multi-organ dysfunctions [2]. Mortality due to septic shock has decreased in the past decade thanks to the prompt initiation of appropriate antimicrobial therapy [10] and sophisticated organ support therapies such as extracorporeal membrane oxygenation (ECMO), continuous hemodiafiltration (CHDF), and direct hemoperfusion with immobilized polymyxin B fiber (PMX-DHP) [11,12]. Nevertheless, mortality due to septic shock remains high at approximately 20% [13,14,15]. Gram-negative bacteremia induce a more severe inflammatory response in septic patients than Gram-positive bacteremia [16]. Therefore, Gram-negative sepsis is prone to progress to septic shock, and the mortality due to Gram-negative septic shock is still higher than that associated with Gram-positive sepsis at approximately 30–40% [16,17,18]. Aging is also a risk factor for colonization by Gram-negative bacteria, such as the *Klebsiella*, *Escherichia coli*, and *Enterobacter* species, predisposing elderly patients to Gram-negative bacterial sepsis and septic shock [19]. Taken together, these findings indicate that elderly patients are more susceptible to sepsis than younger ones, in particular Gram-negative sepsis, and may suffer sepsis-induced multi-organ dysfunctions, thereby leading to a higher mortality [2,19,20,21].

## 4. Pivotal Role of Tumor Necrosis Factor-α (TNF) in Human Endotoxin Shock

Gram-negative bacteria contain endotoxin, a main component of their bacterial envelope, that is also known as lipopolysaccharide (LPS). Endotoxin/LPS is responsible for the pathogenesis of septic shock induced by Gram-negative bacterial infections [22,23]. Plasma endotoxin/LPS levels are associated with increased severity of illness, as measured by the indicator of total organ dysfunctions [24]. Endotoxin/LPS binds to toll-like receptor 4 (TLR4) expressed on the cellular membrane of macrophages and stimulates the NF-κB-dependent inflammatory cascade to produce tumor necrosis factor (TNF), a strongly representative proinflammatory cytokine [23]. Though TNF is important for host antibacterial resistance [25,26], the excessive production of TNF by the stimulation of bacterium-derived LPS may typically cause endothelial damage, multi-organ dysfunctions, and septic shock [27]. TNF transmits its signals into the cells through TNF-receptor (TNFR) 1 and 2, which are expressed on the surface of various mammalian cells [28]. TNFR1 contains a cytoplasmic ‘death domain’ that induces cell death of TNFR1-expressing cells [28]. Thus, LPS-induced increased TNF production is likely a major contributor to endotoxin/LPS shock. Indeed, patients with septic shock show high TNF levels in the plasma, and their TNF levels are related to the patients’ outcomes [21,29,30]. In addition, the administration of endotoxin/LPS to healthy volunteers increased the plasma TNF levels in the initial phase, suggesting that TNF causes the subsequent inflammatory cytokine response [31,32,33]. Consistently, the administration of TNF to healthy volunteers resulted in a similar inflammatory reaction in humans to that of endotoxin/LPS administration [31,34]. TNF thus plays pivotal roles in the inflammatory cascade in humans with endotoxemia/Gram-negative sepsis.

## 5. Generalized Shwartzman Reaction as an Experimental Endotoxin Shock Model

Humans are much more sensitive to LPS than other mammals, such as rodents. The administration of 2–4 ng/kg LPS to healthy volunteers has been shown to evoke various inflammatory responses, including the elevation of plasma TNF [31,32,33]. In contrast, even when mice were administered 2–4 μg/kg LPS, which was almost 1000 times the concentration administered to humans, they only showed a slight elevation of plasma TNF without any organ damage [35]. We then considered that induction of a hypersensitive response to LPS was important for simulating/mimicking human endotoxin shock in mice. The generalized Shwartzman reaction is well recognized as a hypersensitive innate immune response in experimental animals [36].

Roughly 90 years ago, Shwartzman first described the intradermal injection of the sterile culture filtrate of the Gram-negative bacterium *Bacillus Salmonella typhosus* as a preparatory injection and noted that the subsequent intravenous injection with a provocative dose of the same culture filtrate 24 h later induced severe hemorrhagic necrosis at the first injection site in rabbits [37]. If the provocative challenge was too short (less than 2 h) or too long (beyond 48 h), dermal necrosis did not occur. Shwartzman also performed the experiment using a similar preparation with sterile culture filtrate of Gram-positive streptococcal species but failed to duplicate dermal reactions, suggesting that the heat-stable component of Gram-negative bacteria (later identified as LPS) is important for inducing this reaction [36]. However, Shwartzman described only the localized dermal response and not the systemic response in this phenomenon. According to the review of Chahin et al., the generalized Shwartzman reaction was actually discovered by Sanarelli four years earlier than the report by Shwartzman [36]. The generalized Shwartzman reaction is now considered a potentially lethal endotoxin shock reaction that can be induced by the administration of a sublethal dose of LPS into LPS-primed animals at an interval of 24 h (Figure 1A) [38].

## 6. Generalized Shwartzman Reaction Induced by Interleukin (IL)-12 Priming and Sublethal LPS Challenge

Though the generalized Shwartzman reaction is well recognized as an endotoxin shock model, it cannot be induced every time in animals, as this phenomenon is likely an example of a related phenomenon known as endotoxin tolerance [35,36,40]. Through the generalized Shwartzman reaction, injection of interleukin (IL)-12 or IFN-γ, instead of LPS, can induce a lethal reaction in mice if they are further challenged with a sublethal dose of LPS [41]. IL-12, which is mainly produced by macrophages, is a Th 1 cytokine that stimulates natural killer (NK) cells, natural killer T (NKT) cells, and T cells to produce IFN-γ [42,43]. Neutralizing antibody against IL-12, when administered together with the priming agent, prevents the lethal reaction in mice primed with either LPS or IL-12 but not with IFN-γ [41]. Thus, IL-12 and its induced IFN-γ in the priming phase are crucial for the occurrence of the generalized Shwartzman reaction (Figure 1B) [41,44]. Upon subsequent LPS challenge, the lethal Shwartzman reaction is induced by the massive production of inflammatory cytokines, particularly TNF, that act on the target organs already sensitized by IFN-γ and resultant multi-organ dysfunctions in mice [41]. TNF is a key cytokine in the generalized Shwartzman reaction as well as endotoxin shock and results in high mortality [39,41,44]. Notably, IL-12 priming rather than LPS priming can constantly induces the lethal generalized Shwartzman reaction in mice [39,44]. We speculate that although the doses of LPS are different from LPS tolerance, two times LPS injections (LPS priming and subsequent challenge with LPS) may induce a LPS tolerant condition (as described later), leading to failure to effectively induce a generalized Shwartzman reaction [35,40].

## 7. Crucial Role of NKT Cells and Their Produced IFN-γ for the Murine Generalized Shwartzman Reaction

In the generalized Shwartzman reaction, which is induced by IL-12 priming and subsequent LPS challenge in mice, depletion of both NK cells and NKT cells (by anti-NK1.1 antibody) greatly reduced the elevation of serum IFN-γ after IL-12 priming and reduced the mortality after LPS challenge, whereas depletion of NK cells alone (by anti-asialo GM1 antibody) only partially decreased the serum IFN-γ and did not affect the mortality [44]. NKT cells and their produced IFN-γ may be crucial in the priming phase for the induction of lethality in the generalized Shwartzman reaction. However, NK cells play important roles in protecting their host against sepsis/bacterial infections [45], as they have potent IFN-γ-producing capability in response to bacterial infection and can effectively induce bacterial killing by macrophages, which is crucial for bacterial elimination [45]. We should not underestimate NK cell-producing IFN-γ even in the generalized Shwartzman reaction as well as sepsis/infections.

## 8. Involvement of Innate Lymphocytes in the Generalized Shwartzman Reaction

The innate immune system serves as the first line of the host defense against invading bacteria. NK cells, NKT cells, and other innate lymphocytes are crucially involved in this front-line defense [46,47]. Severe surgical stress, such as burn injury, drastically reduces the IFN-γ-producing capability of these cells, thereby increasing susceptibility to sepsis/infection [45,48]. In turn, excessive activation of these innate lymphocytes can induce an exaggerated inflammatory response through the substantial production of IFN-γ [45,48]. Thus, dysregulation (including both down- and up-regulation) of these innate lymphocyte functions can be harmful for the host defense [47,48]. In the generalized Shwartzman reaction, these innate lymphocytes also greatly contribute to the lethality of this reaction. The combination of both IFN-γ-producing lymphocytes (in the priming phase) and TNF-producing macrophages (in the subsequent phase) plays a key role in the generalized Shwartzman reaction (Figure 1B).

## 9. Age-Dependent Increase in Mortality Due to the Generalized Shwartzman Reaction in Mice

Interestingly, the generalized Shwartzman reaction induced by IL-12 priming and subsequent low-dose LPS challenge drastically exacerbates the lethality with increasing age in mice (Figure 2A) [39]. We primed young C57BL/6 mice (4 weeks old) with 0.5 μg/body of IL-12, and, 24 h later, we challenged them with 50 μg/body of LPS [39]. At such doses of IL-12 and LPS, all of the young mice survived the generalized Shwartzman reaction (Figure 2A). However, when older mice (beyond 20 weeks old) were primed/challenged with the same doses of IL-12 and LPS, none survived (Figure 2A), despite showing a mean body weight that was markedly heavier than that of young mice [39]. The serum TNF levels at 1 h after LPS challenge (following IL-12 priming) were also age-dependently increased in the generalized Shwartzman reaction (Figure 2B), indicating TNF-induced mortality, although no age-dependent increase in TNF was observed in the mice challenged with LPS alone.

In contrast, the serum IFN-γ levels were not age-dependently increased in mice. However, interestingly mononuclear cells in the liver (but not spleen) showed significantly increased IFN-γ production by IL-12 priming in middle-aged mice (24 weeks old). NK cells and NKT cells are abundant in the liver and have a potent IFN-γ-producing capability in response to IL-12 priming [46]; we therefore consider that the liver plays crucial roles in the generalized Shwartzman reaction [39,44,46]. However, NK and NKT cells do not increase age-dependently [39]. Regarding age-dependent increases in IFN-γ-producing innate lymphocytes, CD8^+^CD122^+^T cells are age-dependently increased and have potent IFN-γ-producing capability, as described later.

## 10. Immunosenescence and Thymus-Independent T Cells in Mice

Age-related remodeling of the immune system is termed ‘immunosenescence’ and profoundly affects changes in the host defense [49,50]. In particular, T cell-mediated immunosenescence may be greatly affected by thymus involution, which drastically curtails the production of thymus-dependent T cells. However, this loss of thymic cell output with aging results in no significant changes to the systemic T cell counts [51]. Reductions in the peripheral T cell numbers in elderly individuals may be compensated by the thymus-independent expansion of mature T cells [46,52]. These thymus-independent T cells have unique characteristics of more primitive lymphocytes, such as intermediate levels of T cell receptor (TCR) expression [46,52,53]. Their TCR intensity is lower than that of regular αβTCR^+^ T cells but higher than that of immature CD4^+^CD8^+^ double-positive thymocytes (TCR dull expression) before selection (Figure 3A) [54] (as described in detail in next paragraph). Though aging also reportedly attenuates the macrophage function [55,56], aged mice showed no marked reduction in cytokine productions [39,57]. It may be possible that age-dependently increased thymus-independent T cells compensates for the reduced macrophage function in the innate immune system in elderly hosts.

## 11. Age-Dependent Increases in Murine CD8^+^CD122^+^ T Cells

Interestingly, these intermediate TCR-expressing cells are positive for CD122 [46], which is an IL-2 receptor β-chain that contributes to the formation of IL-2 and IL-15 signaling complexes, as both cytokines share this receptor subunit (CD122) [58,59]. CD122 is also highly expressed on NK cells [60,61]. Among CD122^+^ cells with intermediate TCR (TCR^int^) in the murine liver, two thirds are NK1.1 marker-positive (NKT cells), and one third are NK1.1-negative (Figure 3B) [62,63]. In these murine NKT cells, two thirds are CD4-positive, but one third are double-negative (CD4^−^ CD8^−^) [46,52]. In contrast, in the CD122^+^NK1.1^−^ cells, two thirds are CD8-positive, but one third are double-negative (Figure 3B) [46,52]. Interestingly, these CD122^+^ NK1.1^−^ CD8^+^ cells, also known as CD8^+^CD122^+^T cells, progressively increase with age and have a potent IFN-γ-producing capability [39,64]. However, there are some caveats, as the antigen-specific memory CD8^+^T cells with a memory phenotype (CD44^int^) are also CD122-positive [65], although their CD122 expression is intermediate and distinguished from CD122^high^ NK cells [59]. Antigen-specific memory CD8^+^T cells (CD44^int^, CD122^int^) are generated/primed by various environmental antigens and long-lived [65], while the current CD8^+^CD122^+^T cells (CD44^high^, CD122^high^) may be generated independently of environmental antigens and are short-lived; these cells are therefore also termed ‘memory phenotype’ CD8^+^T cells [66,67,68]. They are not engaged in chronic responses to environmental antigens but are subject to nonantigen-specific stimulation through contact with cytokines released in response to various stressors, such as bacterial infections and/or LPS [66,67,68].

## 12. Crucial Role of CD8^+^CD122^+^T Cells in Mortality Due to the Generalized Shwartzman Reaction in Aged Mice

By the generalized Shwartzman reaction in middle-aged mice (24 weeks old), depletion of NK and NKT cells did not reduce the elevation of serum IFN-γ after IL-12 priming, whereas additional depletion of CD8^+^CD122^+^T cells to NK/NKT cells (by anti-TMβ1 antibody) markedly reduced the elevation of IFN-γ after IL-12 priming [39]. Consistently, depletion of CD8^+^CD122^+^T cells in addition to NK/NKT cells drastically decreased the mortality due to the generalized Shwartzman reaction in middle-aged mice, accompanied by the marked reduction in serum TNF levels after LPS challenge [39]. Adoptive transfer of CD8^+^CD122^+^T cells from aged mice markedly increased the mortality due to the generalized Shwartzman reaction in young mice, accompanied by increases in IFN-γ after IL-12 priming and in TNF after subsequent LPS challenge [39]. Thus, age-dependent increases in CD8^+^CD122^+^T cells greatly contribute to the high mortality due to the generalized Shwartzman reaction in aged mice (Figure 1C).

## 13. Beneficial Roles of CD8^+^CD122^+^T Cells in Host Defense

To clarify why the numbers of CD8^+^CD122^+^T cells in mice increase with age, we pretreated young mice with IL-15, which induces CD8^+^CD122^+^T cells [57]. As expected, IL-15-induced CD8^+^CD122^+^T cells augmented the generalized Shwartzman reaction, even in young mice, while the depletion of these CD8^+^CD122^+^T cells eliminated this harmful effect of IL-15 [57]. Interestingly, IL-15-induced CD8^+^CD122^+^T cells also increased the survival after lethal bacterial infection (*Escherichia coli*) in young mice with enhanced IFN-γ production. In addition, IL-15-indcued CD8^+^CD122^+^T cells also increased the anti-tumor activity against EL4 cells (murine lymphoma cells) [57]. Consistently, α-galactoceramide, which is a synthetic ligand for NKT cells, induced the proliferation of CD8^+^CD122^+^T cells with an anti-tumor function in mouse liver [69]. Thus, age-dependently increased CD8^+^CD122^+^T cells may augment antibacterial and anti-tumor immune responses [57]. These beneficial effects of CD8^+^CD122^+^T cells on the host defense appear to be rational in aged hosts, as the elderly are susceptible to bacterial infections and tumor progression/invasion. However, there are some questions raised. Gain of CD8^+^CD122^+^T cells may be just affected by cytokines such as IL-15, but not aging itself. While IL-15 levels are also increased in the liver of aged mice in comparison to the young mice [57]. Therefore, aging may possibly affect the gain of CD8^+^CD122^+^T cells via an increase in IL-15 levels. However, phrase of ‘age-dependent immune-alteration’ that we used in this review may be termed as ‘correlation of immunological status in the hosts at a certain age’, because strictly speaking, we just observed the age-correlated alteration of immune system.

## 14. In Vitro Shwartzman Reaction-Like Response in Human Peripheral Blood Mononuclear Cells (PBMCs)

Is such an age-dependent augmentation of the generalized Shwartzman reaction observed in humans? Five decades ago, Starzl et al. reported three patient cases the suggesting Shwartzman reaction after renal transplantation [70]. However, evaluating the Shwartzman reaction in healthy volunteers is ethically unacceptable. Therefore, little information or evidence of the human Shwartzman reaction has been gathered thus far, although humans are much more sensitive to LPS than animals.

We therefore examined the in vitro Shwartzman reaction-like response using human PBMCs including lymphocytes and macrophages [21]. PBMCs obtained from healthy adult volunteers were cultured with IL-12 (20 μg/mL) for 24 h, followed by culture with LPS (10 ng/mL) for 24 h to examine their TNF production (Figure 4A). CD56^+^NK cells, CD56^+^T cells, and CD57^+^T cells produced IFN-γ in response to IL-12 stimulation, and this IFN-γ primed macrophages to produce large amounts of TNF on subsequent LPS stimulation, suggesting that an in vitro Shwartzman reaction-like response was induced in human PBMCs (Figure 4B) [21]. Nevertheless, we should be careful of the differences in the immune systems between the human and murine, because these immune systems are similar in some parts but not exactly same.

## 15. Age-Dependent Augmentation of the In Vitro Shwartzman Reaction-Like Response in Human PBMCs

Interestingly, PBMCs from healthy elderly volunteers (≥70 years old) markedly produced larger amounts of TNF after LPS stimulation following IL-12 stimulation than did PBMCs from younger adult volunteers (20–40 years old) (Figure 5A), suggesting that the in vitro Shwartzman-like reaction was augmented in elderly PMBCs. In contrast, PBMCs from child volunteers showed no enhanced TNF production after IL-12/LPS stimulation (Figure 5A), suggesting that the Shwartzman-like reaction did not occur in child PBMCs [21]. IL-12-induced IFN-γ production from PBMCs was also age-dependently increased. However, there were no marked differences in the TNF production after stimulation with LPS alone (without IL-12 stimulation) among PBMCs from children, adults, and elderly volunteers (Figure 5A). Thus, the age-dependent enhancement of the Shwartzman-like reaction may occur in human PBMCs.

The proportion of CD56^+^NK, CD56^+^T, and CD57^+^T cells with potent IFN-γ-producing capabilities also increased with age in human PBMCs (Figure 5B) [21]. In particular, CD57^+^T cells were markedly increased (more than five-fold) in elderly PBMCs compared with child PBMCs (Figure 5B). These age-dependent increases in innate lymphocytes, which have potent IFN-γ-producing capability, may play a key role in the Shwartzman reaction in humans (Figure 4C). Interestingly, both CD56^+^T and CD57^+^T cells show intermediate TCR intensity similar to murine NKT cells and CD8^+^CD122^+^T cells, which are involved in thymus-independent T cells [71].

## 16. High Mortality and Morbidity of Septic Shock in Elderly Patients Showing Similar Plasma Endotoxin Levels as Adults (Elderly vs. Adults)

We examined the rate of septic shock and its mortality in elderly (≥70 years old) and adult (<70 years old) patients with sepsis/severe sepsis, according to the recent definition [8]. Though both the elderly and adult septic patients showed similar plasma endotoxin levels (7.6 vs. 7.4 pg/mL in average), the elderly patients showed higher serum TNF levels and were more susceptible to septic shock than the younger adult patients, leading to an extremely high mortality due to septic shock compared with younger patients (91% vs. 25%) (Figure 6) [21]. These findings may confirm the characteristic clinical picture in elderly patients with sepsis/septic shock, which have been described before, and are consistent with the experimental results of the age-dependent augmentation of the murine generalized Shwartzman reaction [39] and the in vitro Shwartzman-like reaction using human PBMCs [21].

## 17. Management of Severe Sepsis/Septic Shock in Elderly Patients: Endotoxin Tolerance as a Potential Therapeutic Strategy in the Future

Elderly patients are susceptible to Gram-negative septic shock with enhanced production of TNF, resulting in a high mortality. When considering effective therapies for elderly patients in such a critical condition, it may be important to reduce the inflammatory cytokine responses, particularly the TNF production by macrophages, and augment the bactericidal activity of these macrophages. However, this seems difficult, as both inflammatory cytokine production and bacterial killing are closely associated with macrophage activation and appear to be synchronized phenomena at a glance. For instance, the induction of CD8^+^CD122^+^T cells by IL-15 pretreatment rendered mice susceptible to the generalized Shwartzman reaction (indicating an exaggerated inflammatory response) but resistant to bacterial infection due to deriving an appropriate inflammatory response [57]. Regarding the generalized Shwartzman reaction, aging mainly affects innate lymphocytes and does not directly affect macrophages [21,39].

Interestingly, endotoxin tolerance mainly affects macrophages, drastically reducing the inflammatory cytokine production—particularly TNF production—while strongly augmenting the bactericidal activity [35,40]. These effects can be induced in aged hosts [72,73] and do not directly affect the innate lymphocytes because endotoxin tolerance can be induced in mice even after the elimination of these lymphocytes (our unpublished data) (Figure 7). Though how to clinically induce endotoxin tolerance remains unclear, the mechanisms underlying endotoxin tolerance are quite attractive for the treatment of sepsis in elderly patients, and this therapeutic strategy may become an effective tool for delivering critical care for refractory septic shock in elderly patients.

## 18. Concluding Remarks

Elderly patients are susceptible to septic shock, particularly Gram-negative septic shock (also known as endotoxin shock), and they have an increased associated mortality compared with younger patients. The generalized Shwartzman reaction is well recognized as an experimental endotoxin shock. Aged mice have similarly been shown to be susceptible to the generalized Shwartzman reaction with an increased mortality accompanied by enhanced TNF production. Consistent with this murine model, the in vitro Shwartzman reaction-like response was shown to be age-dependently augmented in human PBMCs, as shown by enhanced TNF production. Interestingly, the age-dependent increase in TCR-intermediate innate lymphocytes, such as CD8^+^CD122^+^T cells for mice and CD57^+^T cells for humans, was found to be closely involved in this age-dependent augmentation of the Shwartzman reaction.

Endotoxin tolerance mainly affects macrophages and drastically reduces their TNF production but strongly activates their bactericidal activity in response to infection. Importantly, endotoxin tolerance can be age-independently induced; therefore, if this tolerant condition can be safely induced in elderly patients, it may be a useful therapeutic strategy for managing refractory septic shock in this population.

## Figures and Tables

**Figure 1 ijms-20-03260-f001:**
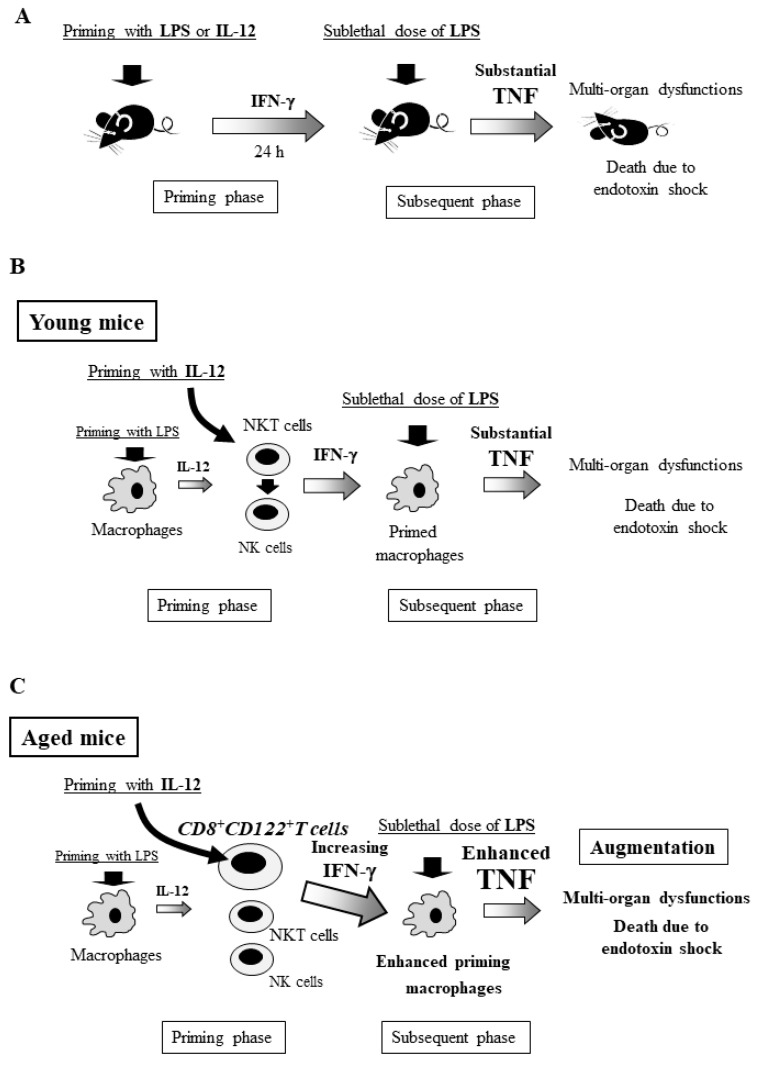
The generalized Shwartzman reaction induced by interleukin (IL)-12 priming and subsequent lipopolysaccharide (LPS) challenge. (**A**) Priming with LPS or IL-12 induces IFN-γ production in mice, and 24 h later, subsequent LPS challenge induces substantial tumor necrosis factor (TNF) production, resulting in the generalized Shwartzman reaction. (**B**) In young mice, IL-12 priming activates natural killer (NK) and natural killer T (NKT) cells to produce IFN-γ, which primes macrophages. Subsequent LPS challenge strongly activates the primed macrophages to produce large amounts of TNF, resulting in the generalized Shwartzman reaction. (**C**) In aged mice, age-dependently increased IL-12-activating CD8^+^CD122^+^T cells augment IFN-γ production, resulting in the further enhancement of TNF production by macrophages and leading to increased lethality of the generalized Shwartzman reaction. Figures are revised from [39].

**Figure 2 ijms-20-03260-f002:**
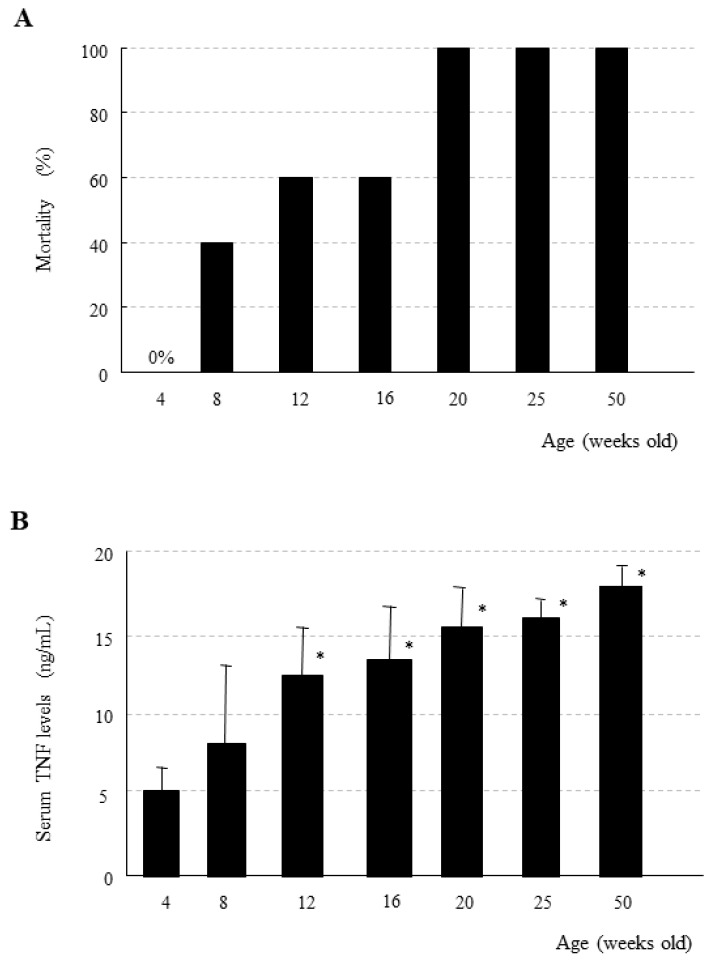
Age-dependent increase in mortality due to the generalized Shwartzman reaction accompanied by elevation of serum TNF. (**A**) Mortality due to the generalized Shwartzman reaction induced by IL-12 priming, and, subsequent LPS challenge was age-dependently increased in mice. (**B**) Serum TNF levels were also age-dependently increased at 1 h after LPS challenge following IL-12 priming. Data are mean (**A**) and means ± SE (**B**) from five mice in each age group. * *p* < 0.01 vs. four weeks old, using an analysis of variance with Scheffe’s F test. Figures are revised from [39].

**Figure 3 ijms-20-03260-f003:**
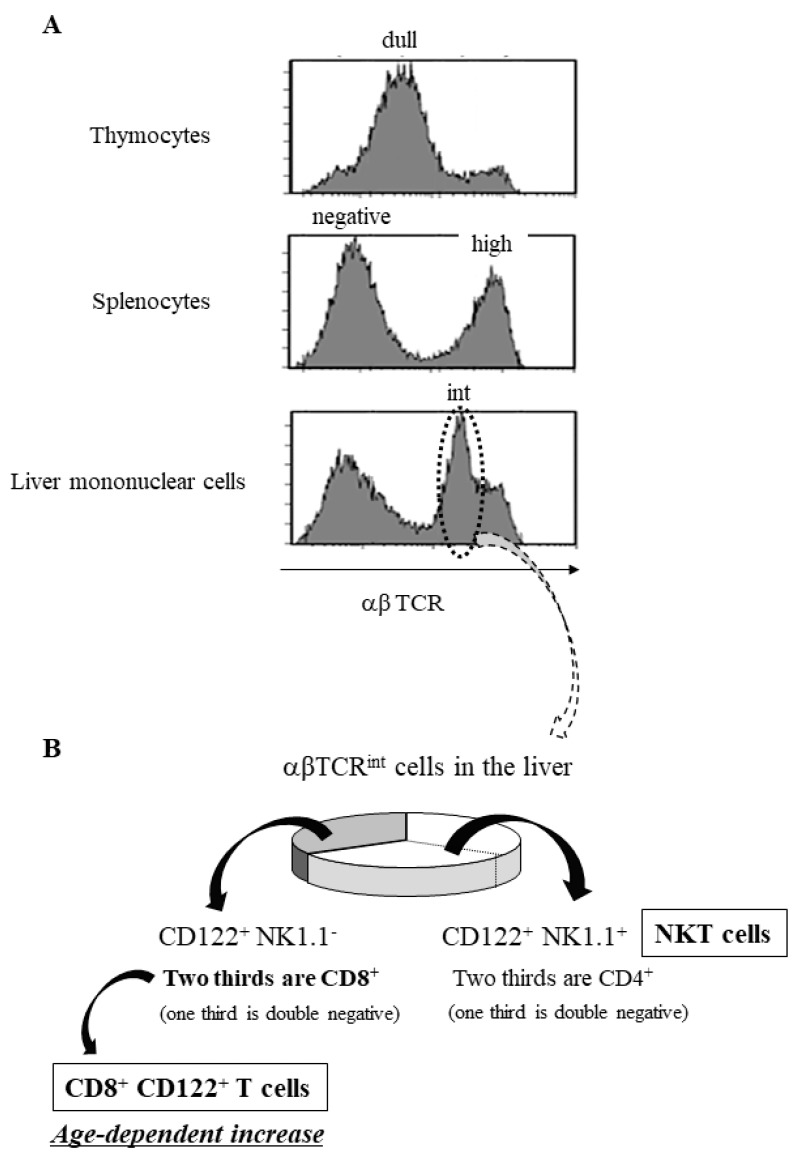
T cell receptor (TCR) intensity in lymphocytes of the thymus, spleen, and liver along with subsets of liver TCR-intermediate cells. (**A**) Immature CD4^+^CD8^+^ double-positive cells show dull TCR intensity in thymocytes, mature T cells show high TCR intensity in the splenocytes, and NKT and CD8^+^CD122^+^T cells show intermediate TCR intensity in the liver mononuclear cells (indicated by dotted circle). (**B**) TCR-intermediate cells in the liver include CD122^+^ NK1.1^+^ cells (NKT cells; two thirds are CD4^+^), and CD122^+^ NK1.1^−^ cells (two thirds are CD8^+^). The cells of this CD8^+^ subset are called CD8^+^CD122^+^T cells and demonstrate an age-dependent increase Figures are revised from [46].

**Figure 4 ijms-20-03260-f004:**
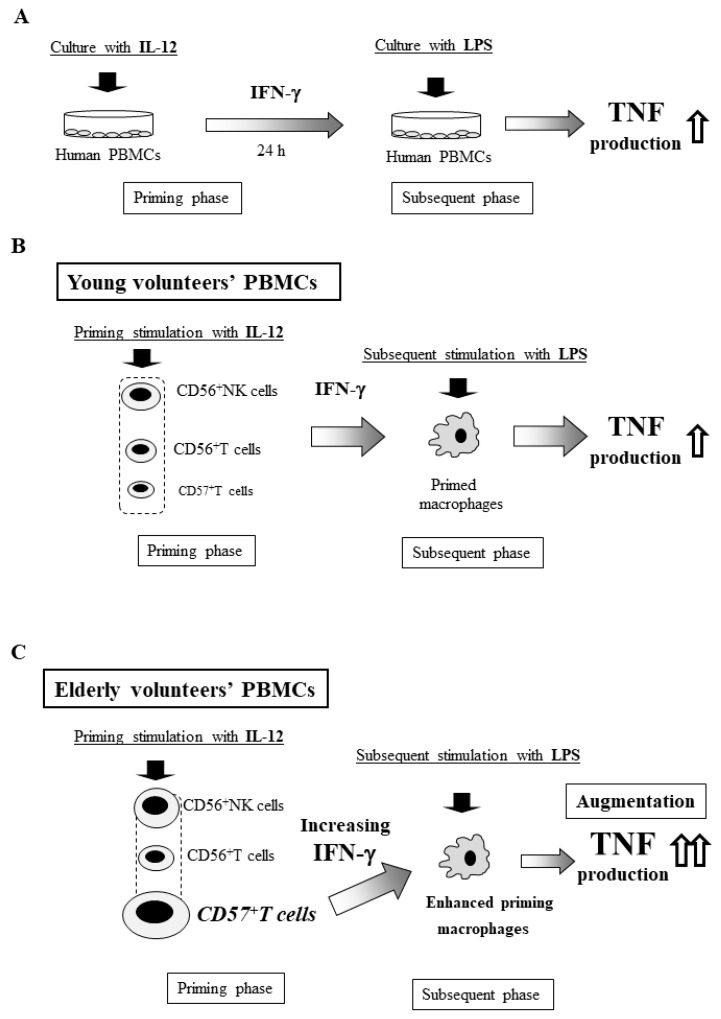
The in vitro Shwartzman reaction-like response in human peripheral blood mononuclear cells (PBMCs). (**A**) IL-12 stimulation and subsequent LPS stimulation 24 h later induced enhanced TNF production in cultured human PBMCs, suggesting an in vitro Shwartzman reaction-like response. (**B**) In PBMCs from young volunteers, IL-12 priming activates CD56^+^NK, CD56^+^T, and CD57^+^T cells to produce IFN-γ, which primes macrophages. Subsequent LPS stimulation strongly activates the primed macrophages to produce large amounts of TNF, suggesting an in vitro Shwartzman reaction-like response. (**C**) In PBMCs from elderly volunteers, IL-12-priming potently activates CD57^+^T cells, which show an age-dependent increase, and augments IFN-γ production, which enhances macrophage priming. The subsequent LPS stimulation further augments the TNF production from potently primed macrophages, suggesting an enhancement of the in vitro Shwartzman reaction-like response. Figures are revised from [21].

**Figure 5 ijms-20-03260-f005:**
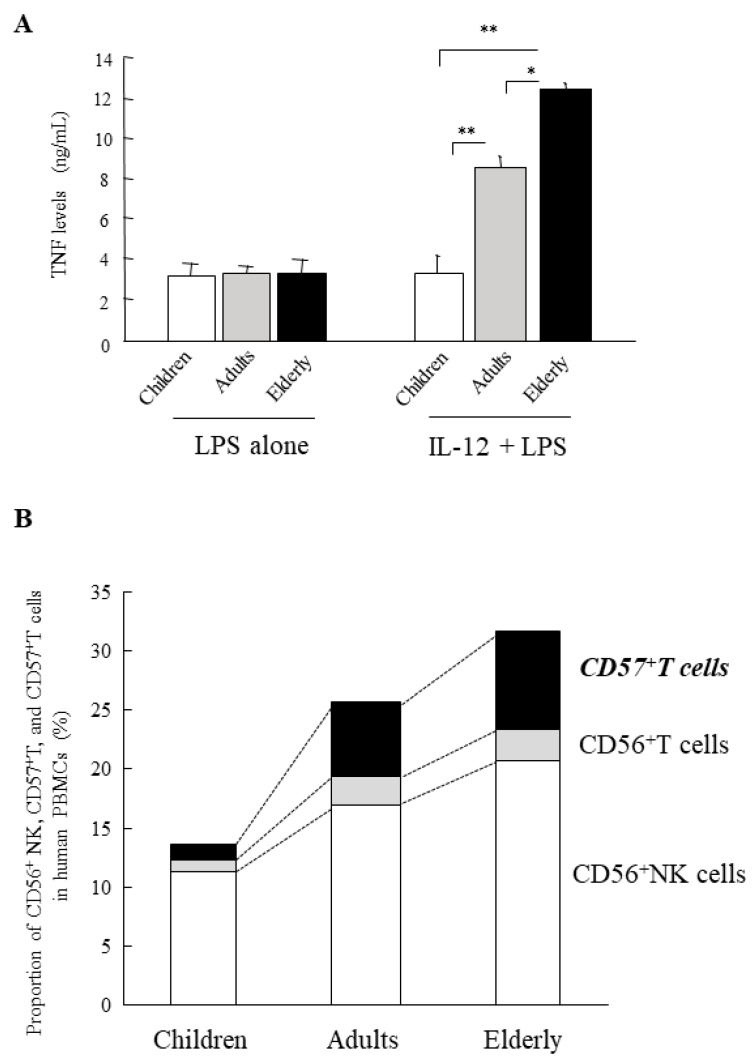
Age-dependent increases in TNF production from human PBMCs by stimulation with IL-12 and LPS and age-dependent increases in CD56^+^NK, CD56^+^T and CD57^+^T cells in human PBMCs. (**A**) Human PBMCs age-dependently increased TNF production by stimulation with LPS following IL-12 stimulation but not by LPS stimulation alone. (**B**) The proportions of CD56^+^NK, CD56^+^T, and CD57^+^T cells were age-dependently increased in human PBMCs. In particular, CD57^+^T cells were markedly increased age-dependently. Data are mean ± SE (**A**) and mean (**B**) from seven, ten, and six healthy volunteers in the child, adult, and elderly groups, respectively. ** *p* < 0.01, * *p* < 0.05, using an analysis of variance with Scheffe’s F test. Figures are revised from [21].

**Figure 6 ijms-20-03260-f006:**
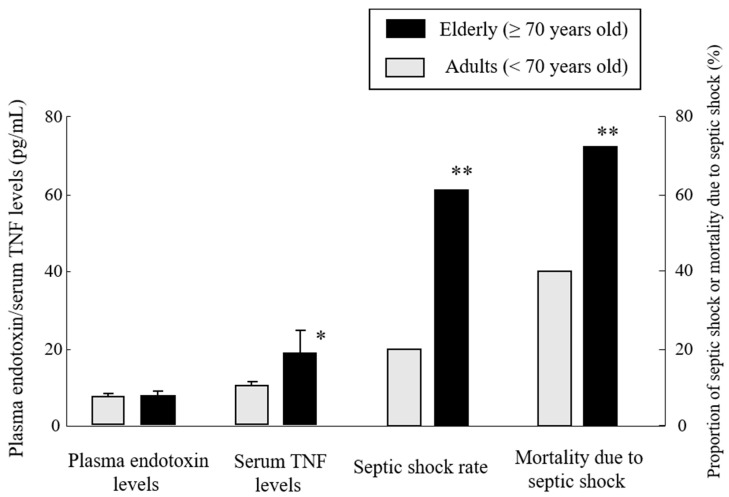
Characteristics of septic shock in elderly patients. Plasma endotoxin levels, serum TNF levels, proportion of septic shock rates, and mortality due to septic shock were compared between elderly and young adult septic patients. Though both patient groups showed similar plasma endotoxin levels, the elderly patients showed markedly higher serum TNF levels and septic shock rates, leading to a high mortality due to septic shock. Data are means ± SE or mean from 18 elderly patients and 20 young adult patients. ** *p* < 0.01, * *p* < 0.05, using the Mann–Whitney *U*-test and chi-squared test. Figure data are revised from [21].

**Figure 7 ijms-20-03260-f007:**
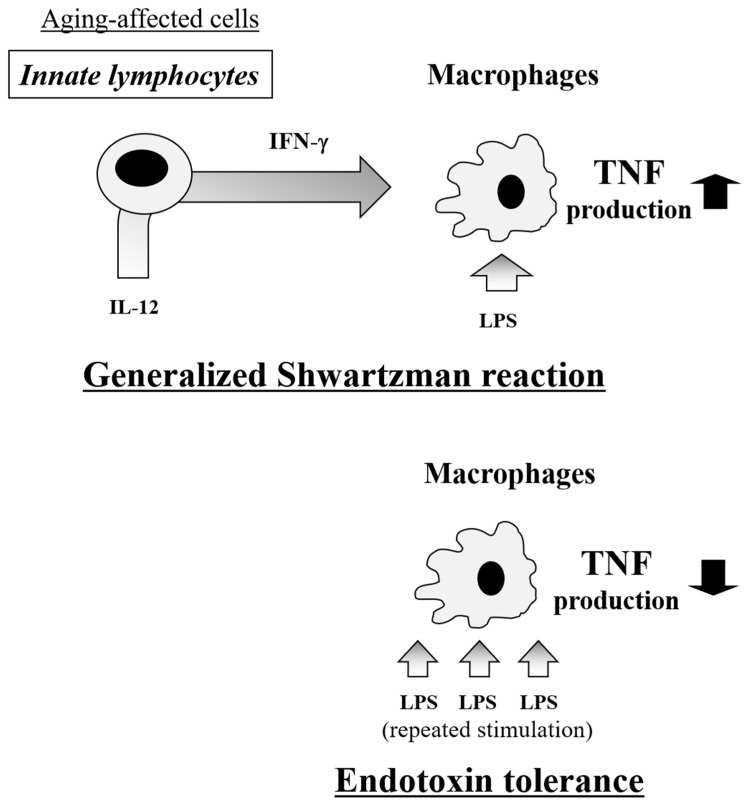
Immune mechanisms in the generalized Shwartzman reaction and endotoxin tolerance. In the generalized Shwartzman reaction, innate lymphocytes, which are age-affected cells, are closely involved in the priming phase to stimulate macrophages, leading to an enhanced TNF production. In contrast, endotoxin tolerance directly affects macrophages to induce a reduction in TNF production.

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
