# Peer review of "Immune Mechanisms Underlying Susceptibility to Endotoxin Shock in Aged Hosts: Implication in Age-Augmented Generalized Shwartzman Reaction"

_ijms, 2019, doi:10.3390/ijms20133260_

Round 1
Reviewer 1 Report
This is an excellent review on immune mechanims of endotoxin shock with respect to age.
The paper can be published essentiallyas it is, I have only some minor comments:
. Page 2, chapter 4: In this chapter, the important clinical studies of Opal and co-workers should be cited, because the authors have given LPS concentrations for septic patients, separately for survivors and non- survivors.
Page 15, Figure 7: Please write Generalized, not generalized in the figure.
Ref. 37: The title ,is STUDIES..... rather than Studies... Please correct
Author Response
To reviewer 1
This is an excellent review on immune mechanisms of endotoxin shock with respect to age. The paper can be published essentially as it is, I have only some minor comments:
1. Page 2, chapter 4: In this chapter, the important clinical studies of Opal and co-workers should be cited, because the authors have given LPS concentrations for septic patients, separately for survivors and non- survivors.
Response: Thank you for your suggestion. Yes, this paper by Dr. Opal SM et al. in Journal of Infectious Diseases is quite important and we think that it is a milestone paper on endotoxin and sepsis/septic shock. We added this paper as ref. #22.
2. Page 15, Figure 7: Please write Generalized, not generalized in the figure.
Response: We corrected to ‘Generalized……’ in Figure 7.
3. Ref. 37: The title ,is STUDIES..... rather than Studies... Please correct.
Response: We corrected the title of ref. #37 to ‘Studies on bacillus…’
Reviewer 2 Report
Undoubtedly, sepsis and septic shock caused by Gram-negative bacteria are among major concerns of health care providers and the population of the elderly is particularly vulnerable. The observed demographic changes in many countries, with a populations living longer and the number of the elderly increasing, imply there is a need for both rapid detection of sepsis and development of effective therapies targeted at the age-defined immunological status of the elderly.
In general all the mechanisms involved in the interactions of endotoxins with hosts are complex and difficult to dissect. Therefore a Review concerning the general Shwartzman reaction and the enhancements of the observed effects as a tool to describe mechanisms of susceptibility of host populations with defined age seems justified. It could provide a valuable aggregation of some scattered observations in animal models and whether they can be and to what extent applied to humans. There are two major concerns that I have with regard to the presentation of the aggregated data in the Review.
Firstly,throughout the paper the Authors have a tendency to address the elderly human population providing observations done on mice. The observations seem relevant, however, one has to be extremely cautious when reporting on certain immunological properties of murine and human immunological systems. I would strongly recommend , wherever possible, to clearly separate the presented observations of age-related immunological properties using mice and human PBM cells, respectively. To what extent the mice data could be translated into the deeper understanding of the elderly human hosts susceptibility to endotoxin?
Secondly, in the text the Authors use the term „age-dependent” to describe the reported correlation of the host age and the corresponding immunological status, manifested as the intensity of the general Shwartzman reaction, a change of the cytokines level, and a variation in the number of specific cells, etc. In my view, this is not in fact an age dependence, but rather a mere correlation of the immunological status of the host at certain age. This status can still be affected by cytokines, an thus not directly related to age (see paragraph 13. induction of CD22+CD122+T cells with IL-15). A brief explanation of the ‘age-dependence’ should be added.
My specific comments:
Line 45: The paragraph title should be shortened to read „2. Definition of septic shock” as the evolution of what constitutes a septic shock is presented.
Line 73: Please re-write: „Mortality due to to septic shock has decreased”.
Line 87: LPS (endotoxin) is the main component of the Gram-negative bacterial envelope.
Line 94: […] LPS may … . It typically does endothelial damage, etc.
Line 142: „[…] it cannot be induced at just any point in animal ...”The sentence should be re-written, re-worded.
Line 187: Are these original data? A reference should be provided.
Line 222: „We believe, [...]„ Anything to substantiate this claim?
Figures 3 and 4: References should be added.
Line 331: The title of the paragraph should be re-worded. […] similar plasma endotoxin levels as adults (elderly vs. adults?).
Figure 6: For the sake of clarity the 3D graphs should be avoided (Is 61% bar below the 60% line?).
Line 368: Is the previously unpublished material allowed in this type of publication (i.e. a Review)?
References 23 and 45: The citation details are missing/incomplete.
p { margin-bottom: 0.25cm; line-height: 115%; background: transparent none repeat scroll 0% 0%; }
Author Response
To reviewer 2
Undoubtedly, sepsis and septic shock caused by Gram-negative bacteria are among major concerns of health care providers and the population of the elderly is particularly vulnerable. The observed demographic changes in many countries, with populations living longer and the number of the elderly increasing, imply there is a need for both rapid detection of sepsis and development of effective therapies targeted at the age-defined immunological status of the elderly.
In general, all the mechanisms involved in the interactions of endotoxins with hosts are complex and difficult to dissect. Therefore, a Review concerning the general Shwartzman reaction and the enhancements of the observed effects as a tool to describe mechanisms of susceptibility of host populations with defined age seems justified. It could provide a valuable aggregation of some scattered observations in animal models and whether they can be and to what extent applied to humans. There are two major concerns that I have with regard to the presentation of the aggregated data in the Review.
1. Firstly, throughout the paper the Authors have a tendency to address the elderly human population providing observations done on mice. The observations seem relevant, however, one has to be extremely cautious when reporting on certain immunological properties of murine and human immunological systems. I would strongly recommend, wherever possible, to clearly separate the presented observations of age-related immunological properties using mice and human PBM cells, respectively. To what extent the mice data could be translated into the deeper understanding of the elderly human hosts susceptibility to endotoxin?
Response: We agree that the rapid detection of sepsis and development of effective therapies targeting the age-defined immunological status of the elderly are quite important, as the aging population is rapidly increasing in many countries. Although the human immune system is not completely the same as that in mice, we believe that animal studies hold some clues to clarifying the mechanisms underlying why elderly humans are susceptible to endotoxins. As we have no effective tools/means with which to explore these mechanisms other than animal studies at present, we should bear in mind the differences in the immune systems between humans and mice.
We have therefore added the following sentences to the end of paragraph 14:
“Nevertheless, we should be wary of differences in the immune system between humans and mice, as these systems are similar in some respects but not exactly the same.’
2. Secondly, in the text the authors use the term “age-dependent” to describe the reported correlation of the host age and the corresponding immunological status, manifested as the intensity of the general Shwartzman reaction, a change of the cytokines level, and a variation in the number of specific cells, etc. In my view, this is not in fact an age dependence, but rather a mere correlation of the immunological status of the host at certain age. This status can still be affected by cytokines, an thus not directly related to age (see paragraph 13. induction of CD8+CD122+T cells with IL-15). A brief explanation of the ‘age-dependence’ should be added.
Response: Strictly speaking, we merely observed age-correlated changes in the immune system, as per your comments. CD8+CD122+T cells can be induced by IL-15 in the liver of young mice. However, we also confirmed that IL-15 levels are increased in the aged liver. Therefore, aging may affect the gain of murine CD8+CD122+T cells via an increase in IL-15 levels. We have therefore added the following text to the end of paragraph 13:
‘However, several questions remain. The gain of CD8+CD122+T cells may simply be affected by cytokines, such as IL-15, rather than aging itself. Of note, IL-15 levels are also increased in the liver of aged mice compared to young mice [57]. Therefore, aging may affect the gain of CD8+CD122+T cells via an increase in IL-15 levels. The phrase ‘age-dependent immune-alteration’ used in this review may thus be taken to refer to the ‘correlation of the immunological status in the hosts at a certain age’, as strictly speaking, we simply observed age-correlated changes in the immune system.’
Specific comments:
3. Line 45: The paragraph title should be shortened to read “2. Definition of septic shock” as the evolution of what constitutes a septic shock is presented.
Response: We changed the paragraph title as your comment.
4. Line 73: Please re-write: “Mortality due to septic shock has decreased”.
Response: We changed to ‘decreased’.
5. Line 87: LPS (endotoxin) is the main component of the Gram-negative bacterial envelope.
Response: We changed to ‘envelope’.
6. Line 94: […] LPS may … . It typically does endothelial damage, etc.
Response: We changed to ‘typically’.
7. Line 142: „[…] it cannot be induced at just any point in animal ...”The sentence should be re-written, re-worded.
Response: We have now changed this to ‘it cannot be induced consistently in animals, ’.
8. Line 187: Are these original data? A reference should be provided.
Response: It is our previous experimental data (ref. #39). Then we added that reference [39]. Thank you.
9. Line 222: „We believe, [...]„ Anything to substantiate this claim?
Response: We have now changed this to ‘The age-dependently increased thymus-independent T cells may compensate…’.
10. Figures 3 and 4: References should be added.
Response: We added the ref. #46 in Figure 3 and the ref. #21 in Figure 4. We also added the ref. #39 in Figure 1.
11. Line 331: The title of the paragraph should be re-worded. […] similar plasma endotoxin levels as adults (elderly vs. adults?).
Response: We changed the sentence, ‘…. similar plasma endotoxin levels as adult (elderly vs. adult)’. Thank you.
12. Figure 6: For the sake of clarity the 3D graphs should be avoided (Is 61% bar below the 60% line?).
Response: We changed Figure 6 to 2D graphs.
13. Line 368: Is the previously unpublished material allowed in this type of publication (i.e. a Review)?
Response: We believe that such inclusion is permitted. We as well as other investigators sometime use unpublished data/observations in review articles. We are also preparing our next paper using these data. However, if the Int. J. Mol. Sci. refutes this claim, we will delete this description.
14. References 23 and 45: The citation details are missing/incomplete.
Response: We deleted these references. Thank you.
15. p { margin-bottom: 6.25px; line-height: 115%; background: transparent none repeat scroll 0% 0%; }
Response: We used the attached template and corrected. Thank you.